# GiRAFR improves gRNA detection and annotation in single-cell CRISPR screens

Qian Yu[1], Paulien Van Minsel[1], Eva Galle [1] & Bernard Thienpont [1,2 ✉]

Novel methods that combine single cell RNA-seq with CRISPR screens enable high-throughput characterization of transcriptional changes caused by genetic perturbations. Dedicated software is however lacking to annotate CRISPR guide RNA (gRNA) libraries and associate them with single cell transcriptomes. Here, we describe a CRISPR droplet sequencing (CROP-seq) dataset. During analysis, we observed that the most commonly used method fails to detect mutant gRNAs. We therefore developed a python tool to identify and characterize intact and mutant gRNAs, called GiRAFR. We show that mutant gRNAs are dysfunctional, and failure to detect and annotate them leads to an inflated estimate of the number of untransformed cells, attenuated downregulation of target genes, as well as an underestimated multiplet frequency. These findings are mirrored in publicly available datasets, where we find that up to 35% of cells are transduced with a mutant gRNA. Applying GiRAFR hence stands to improve the annotation and quality of single cell CRISPR screens.

[1] Laboratory for Functional Epigenetics, Department of Human Genetics, KU Leuven, 3000 Leuven, Belgium. [2] Leuven Institute for Single Cell Omics, KU Leuven, 3000 Leuven, Belgium. ✉email: bernard.thienpont@kuleuven.be

The advent of single-cell omics technologies is revolutionizing the cataloging of cell types and states in development, physiology, and disease, as we recently reviewed[1]. While such studies are mostly descriptive and correlative, recent method developments now also enable functional analysis at scale, with the impact of a multitude of gene perturbations for the first time assessed in parallel on a transcriptome-wide scale. Examples include Perturb-seq, Direct-Capture Perturb-seq, CROP-seq, ECCITE-seq, and others[2–5]. Screens initially encompassed dozens of parallel perturbations, but more recent iterations profiled numbers that are several orders of magnitude higher, extending to the whole transcriptome[6–8]. These screens rely on CRISPR-based systems where guide RNAs (gRNA) are used to target Cas9 to loci of interest for gene knockout, inactivation, or overexpression. By concomitantly reading out a single cell's mRNA and gRNA expression, they enable high-through characterization of genetic perturbation phenotypes. A dedicated data process pipeline that caters to these analyses is however still lacking, and quality control metrics for such experiments have yet to be established.

Here, we describe GiRAFR (Guide RNA Anomaly and Functionality Revealer), a tool based on pysam (https://github.com/pysam-developers/pysam) to perform quality control of single cell CRISPR screens, and to assign gRNAs to cells in a sensitive, mutation-aware manner. GiRAFR enables profiling of gRNA sequence variations, as well as allows pinpointing the sources of this variation, such as induced during library preparation or during virus preparation. In a separate analysis mode, it also detects CRISPR-cas9-induced DNA editing in transcriptome data. By applying GiRAFR to 1 new and 11 publicly available datasets (26 experiments in total[3, 4, 6–14]), we show that mutant gRNA molecule are common in gRNA libraries and that, as a consequence, cells are often inaccurately annotated, and we propose minimal quality metrics for single cell gRNA sequencing libraries. Together, GiRAFR forms a toolbox to analyze single-cell CRISPR screens in a more accurate, reliable, variance-aware, and efficient manner.

## Results

**gRNA mutations prevent gRNA detection.** We transformed A549 cells expressing a tamoxifen-inducible Cas9 with a lentiviral pool for expression of 120 gRNAs (Supplementary Table 1) at low multiplicity of infection. After stringent selection using puromycin and 2 days of Cas9 induction, cells were allowed to grow for another 5 days. Next, 5744 cells were analyzed by CROP-seq as described. Downstream analysis confirmed this experiment to be successful, as cells carrying specific gRNAs showed downregulated expression of gRNA target genes (Fig. 1a, Supplementary Fig. 1a). To further validate functionality, we searched for Cas9-induced indels at gRNA target regions (see methods). This also confirmed our experiment performed as anticipated, since indels were readily identified in highly expressed genes when the gRNA target region was recovered in the transcriptome library (Supplementary Fig. 1b). Surprisingly however, no gRNA was detected in 40% (2317) cells using the established gRNA detection and annotation pipeline, Cell Ranger[15] (Supplementary Fig. 1c). Single-cell CRISPR screens are contingent on successfully and accurately associating a cell's transcriptome with the gRNA it expresses. Similar to what we observed here, however, in every study published thus far, no gRNA is detected in a subset of cells[6–9, 13]. By most researchers, this is attributed to a lack of complete selection for gRNA-transformed cells, leading to the inclusion of non-transformed cells in the analysis, or to an insufficiently high gRNA expression or sequencing depth. In our experiment however, sequencing depth was high, with a

saturation estimated at 98.9%, and gRNA expression in gRNA-positive cells was high, with on average 53 unique molecular identifiers (UMIs) per gRNA per cell (Supplementary Fig. 1d). Remarkably, gRNA-negative cells also expressed the puromycin resistance cassette we used as selection marker, suggesting that these cells were also successfully transformed and selected (Fig. 1b). Closer inspection of reads mapping to the gRNA-expressing plasmid sequence moreover revealed that also gRNA-derived reads are mapped to this region. These however show imperfect mapping, suggesting the presence of gRNA mutations (Fig. 1c).

**Detecting and annotating gRNA mutations.** We therefore set out to develop a tool to detect and annotate mutant gRNAs. This is of importance for several reasons. Firstly, to control the quality of the experimental work preceding gRNA detection, including gRNA oligonucleotide synthesis, gRNA cloning, transformation of cells, and selection of successfully transformed cells. Secondly, single-cell CRISPR screens often rely on gRNAs for accurate discrimination between single cells, having one gRNA, and multiplets, where two or more cells are inadvertently captured together causing them to share barcodes. Such multiplets (as well as double-transformed cells) are characterized by the detection of more than one gRNA, and they should typically be discarded for analysis as the corresponding transcriptome no longer reflects the impact of the CRISPR perturbation. Errors in gRNA detection compromise annotation of these multiplets. And finally, expression of a mutated and potentially dysfunctional gRNA will provoke attenuation or absence of phenotype, thus clouding downstream analyses. Matters are further aggravated by tolerating mismatches for gRNA detection, with the state-of-art feature barcoding pipeline (Cell Ranger) classifying gRNAs with 1 Hamming distance from the designed gRNA as intact.

To remedy these three issues, we developed GiRAFR, a tool to identify mutations in the gRNA expression library and assign intact and mutant gRNAs to cells in single-cell screens. It includes multiple controllable filtering and model fitting parameters to establish accurate spacer calling and provides annotations on both intact and mutant gRNAs (Fig. 1d). Specifically, it first calls a consensus gRNA sequence for each UMI. In doing so, sequencing errors can be filtered from the gRNA pool as detection of multiple reads supporting the same gRNA sequence becomes a prerequisite for gRNA mutation calling (Fig. 1d). Next, it generates a count matrix per detected gRNA, whilst discarding UMIs with fewer than 2 reads to avoid including sequencing errors as consensus gRNAs. Next, by aligning the consensus gRNA sequences to the predefined gRNA library, mutant and intact gRNAs can be identified and mutations can be annotated. Here, we assembled a library of in total 1997 different gRNAs (113 wild-type and 1884 mutant). Once this listing of consensus gRNA sequences was constructed, gRNAs were assigned to cells using the corresponding cell barcode. In each cell, we detected between 0 to over 40 different gRNAs, versus 0 to 22 gRNAs for Cell Ranger (Fig. 1e, Supplementary Fig. 1e). In each cell we detected on average 43 UMIs from 7 different gRNAs, either intact or mutant.

**Potential sources of gRNA mutations.** We next sought to further analyze the sources of these mutant gRNAs, using the GiRAFR output (Supplementary Fig. 1f). Most UMIs that associate with a mutant gRNA (97.3%) originated from 445 unique mutant gRNAs. These mutant gRNAs were each detected in multiple cells (3173 cells in total). They were thus most likely already amplified in the virus pool used for transduction. We cannot discriminate between those originating from inaccurate oligonucleotide synthesis and from errors introduced during gRNA cloning (Supplementary

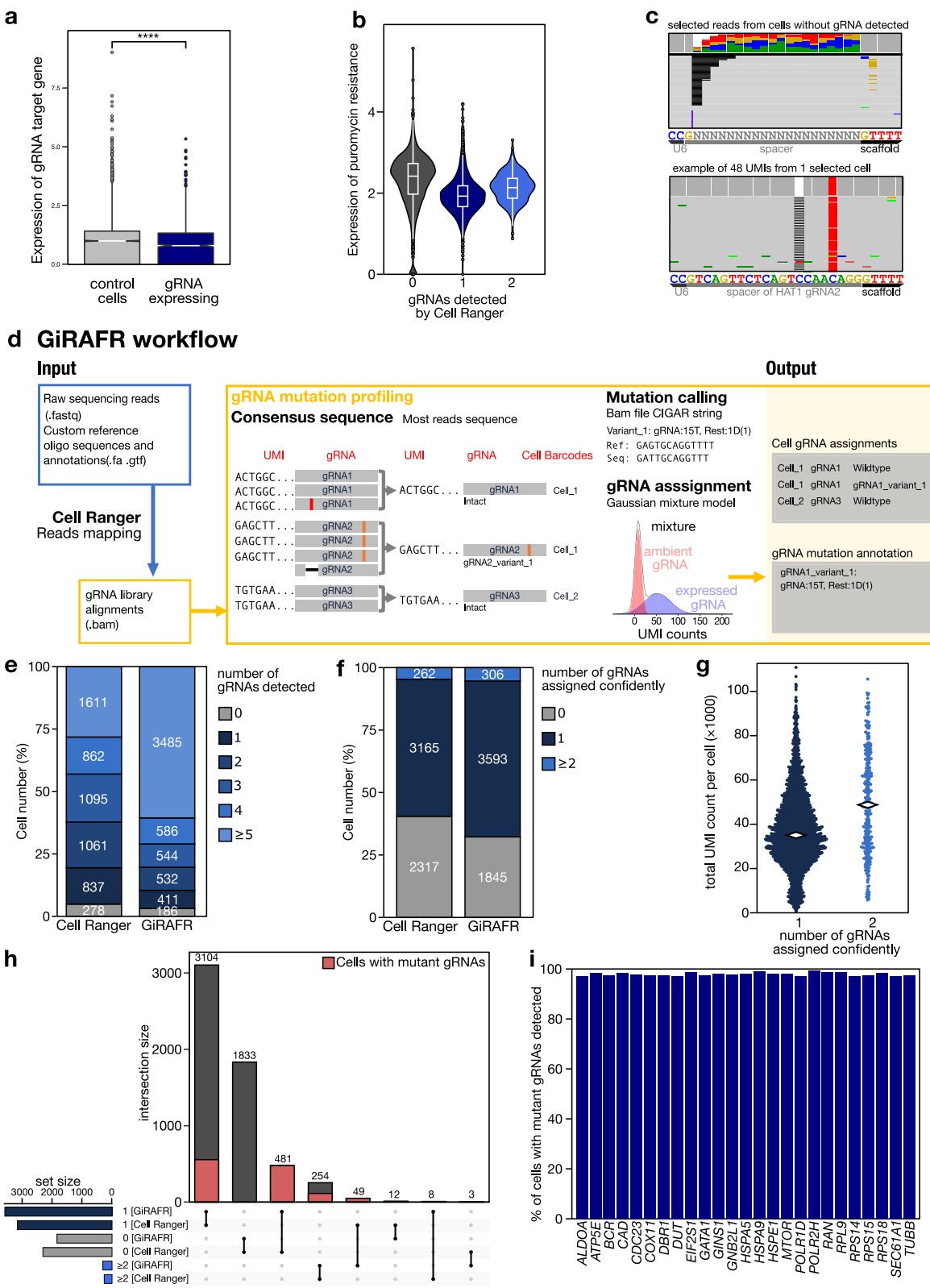

Fig. 1f, scenario 1). A smaller number of mutant gRNAs only appeared in one cell but were supported by all UMIs for that gRNA in that cell (n = 23 gRNAs, 23 cells). These mutations are likely derived from errors during lentivirus transduction, but they could equally represent a rarer mutant gRNA sub-clone present in the virus pool (Supplementary Fig. 1f, scenario 1 or 2). Interestingly, in 1100 cells, both the mutant and intact version of the same gRNA

were encountered. In 1062 of these 1100 cells, only one single UMI supported the presence of the mutant gRNA (validated across multiple reads) while on average 56 UMIs supported the presence of the intact gRNA. Here, the mutation most likely occurred in cDNA preparation (Supplementary Fig. 1f, scenario 4). This contrasts with the cells where >1 UMI supported the detection of both the mutant and intact gRNA (n = 78). Under the latter

**Fig. 1 Development of GiRAFR and its application to an in-house CROP-seq data with comparison of results from Cell Ranger feature barcoding analysis. a** Expression (log-transformed read counts) of gRNA target genes, normalized to their expression in cells with non-targeting gRNAs (control cells). Shown is the aggregate expression of the 14 target genes which show expression in at least 50% of all control cells. ****:$P < 0.0001$ by two-sided $t$ test. Box plot denotes quartile range (box), median (center line), and 1.5× interquartile range (whiskers). **b** Expression of the Puromycin resistance cassette, after mapping the scRNA-seq library to a reference augmented to include the Puromycin resistance cassette. Shown is the log-normalized expression in cells where 0, 1, or 2 gRNAs were detected by Cell Ranger feature barcoding analysis. Box plots inside violin plots denote quartile ranges (box), median (center mark), and 1.5× interquartile range (whiskers). **c** Reads with unique UMIs showing partial mapping to the 20 bp spacer region, using either Ns as a reference (left), or from 1 cell expressing a variant HAT1 gRNA using this gRNA as a reference. **d** Schematic workflow of the GiRAFR pipeline. See methods for additional details. **e**, **f** Number of cells with gRNA spacer assigned by Cell Ranger and GiRAFR (**e**) in the raw count matrix and (**f**) after application of the Gaussian Mixture model represented in panel **d**. **g** Beeswarm plot showing the number of UMIs per cell in cells with single gRNA and 2 gRNAs from GiRAFR. Median numbers are indicated on plot. **h** Comparison of gRNA assignment to cells between Cell Ranger and GiRAFR[25]. The red box indicates cells containing 1 or more mutant gRNAs. **i** Detection rate of mutant gRNAs by GiRAFR in a Perturb-seq dataset[16], containing 103 predesigned mutant gRNAs.

constellation, the mutation was most likely introduced during lentiviral integration, which is associated with a duplication of the gRNA expression cassette[4]. Mutations introduced during this integration/duplication will give rise to an intact and a mutant gRNA both being expressed in a single cell (Supplementary Fig. 1f, scenario 3). Note that 40 cells contain different gRNA mutations, either supported by 1 UMI or by >1 UMI. In a final scenario, a mutation is detected in a single read of a UMI, but not in the other reads of that UMI. This is a sequencing error (Supplementary Fig. 1f, scenario 5) which, as described higher, is filtered out by GiRAFR during the consensus sequence generation step. Here, 27% of the raw mutant molecules were filtered out and 91% of the consensus sequences were constructed from sequence variations. After this filtering, mutant gRNA molecules were widely present in the gRNA library (28.4%). The distribution of mutant gRNA reads showed a bi-model pattern, similar to the wild-type gRNAs, further confirming that they were not sequencing errors (Supplementary Fig. 1g).

**Assigning gRNAs to cells**. Irrespective of the source of mutations, deciding which gRNAs are truly expressed in a cell and which represent artifacts, is a common problem in single-cell CRISPR screen analysis. In GiRAFR, we offer two modes of setting thresholds: one uses fixed UMI thresholds to assign gRNAs to cells, the other one implements the two-components Gaussian mixture model also used by Cell Ranger, to determine dynamic UMI thresholds per gRNA per cell (Fig. 1d). In the latter, default implementation, we included both mutant and intact gRNA UMI counts to model the distribution, implying that different thresholds may be proposed in GiRAFR *versus* Cell Ranger. Importantly, GiRAFR detects more cells with single gRNA (Fig. 1f). Cells with 2 gRNAs show higher total UMI counts, implying that many of these are multiplets (Fig. 1g). When assigning gRNAs to cells using GiRAFR, 481 of the 2317 cells where no gRNA was previously found, appeared transformed with a mutant gRNA, 49 cells where previously only one gRNA was found were in fact multiplets, and 605 cells annotated as having single intact gRNA in fact expressed a mutant gRNA (Fig. 1h). In total 1198 of all 5744 cells included in the analysis (20.7%) contain a mutant gRNA.

To further validate the accuracy and sensitivity of gRNA assignment, we applied GiRAFR to a Perturb-seq dataset where 25 genes were targeted using both intact ($n = 25$) and mutant ($n = 103$) gRNAs[16]. Reassuringly, when solely supplied with 25 intact gRNA sequences, GiRAFR identified 253 mutant gRNAs, including all 103 gRNAs present by design. Of the 14,113 single cells originally annotated as expressing a predesigned mutant gRNA, GiRAFR assigned 98.1% concordantly (Fig. 1i).

**gRNA mutation characteristics and impact**. A large portion of cells discordant between Cell Ranger and GiRAFR express a

mutant gRNA (Fig. 1h). A key question is if gRNA mutations affect their functions. The most frequent mutation is the deletion of thymidine in the TTTT tetramer of the gRNA lower stem structure, but mutations occur both in the gRNA promoter, spacer, and scaffold, and could thus compromise gRNA expression, DNA target recognition, and Cas9 binding, respectively[17] (Fig. 2a).

Therefore, to evaluate the functionality of both intact and mutant gRNAs, we compared the transcript level of gRNA target genes between non-perturbed cells and cells perturbed with a mutant or intact gRNA (Fig. 2b, Supplementary Fig. 1h). gRNAs with a mutant spacer and scaffold failed to reduce expression, and intact gRNAs significantly reduced target gene expression ($P = 8.2 \times 10^{-6}$). Mutant gRNAs thus appear to be dysfunctional. Such an attenuated functionality of mutant gRNAs matches the previously reported reduction in functionality observed for gRNAs with a predesigned mutation[16]. Reanalysing these data confirms the strong CRISPRi-mediated reduction in expression for cells with an intact gRNA, and an attenuated reduction for gRNAs with a single (predesigned or GiRAFR-annotated) mutation in the spacer. In contrast, mutations in the scaffold (detected by GiRAFR) failed to reduce target gene expression, and gRNAs with 2 or more mutations showed a near-normal expression of their target genes (Fig. 2c). gRNA mutations thus have a graded impact on their functionality, depending both on the number and position of the mutation.

Mutant gRNA detection is also needed for the identification of doublets or multiplets. Indeed, when mutant gRNAs abound, droplets can often contain a mutant and an intact gRNA. Undetected cell doublets will however mask the impact of the intact gRNA. This is illustrated by comparing gRNA target gene expression between droplets annotated as containing 1 (intact) gRNA by Cell Ranger but where a second mutant gRNA is detected by GiRAFR. These multiplets reported by GiRAFR showed an attenuated target gene downregulation after perturbation, further emphasizing the importance of detecting mutant gRNAs (Fig. 2b, c, Supplementary Fig. 1i, j).

**gRNA mutations are pervasive**. To assess if gRNA mutations also affect other, independent experiments, we applied GiRAFR to 26 single-cell CRISPR experiments from 11 studies (Supplementary Table 2). While the gRNA library in each study was constructed using a different strategy, resultant sequences usually contain both a region before the spacer (i.e. the end of U6 promoter or the TSO), the spacer region itself that contains the unique gRNA sequence (~G + 20 bp), and the remainder, representing gRNA scaffold and/or capture sequence. GiRAFR detected mutant gRNA molecules in all studies analyzed, at frequencies between 2 and 35%. Different samples from the same study showing concordant results (Fig. 2d), but across studies we observed different mutation spectra and frequencies, ranging from 0.4 to 6.5 mutations per 1000 nucleotides of each gRNA

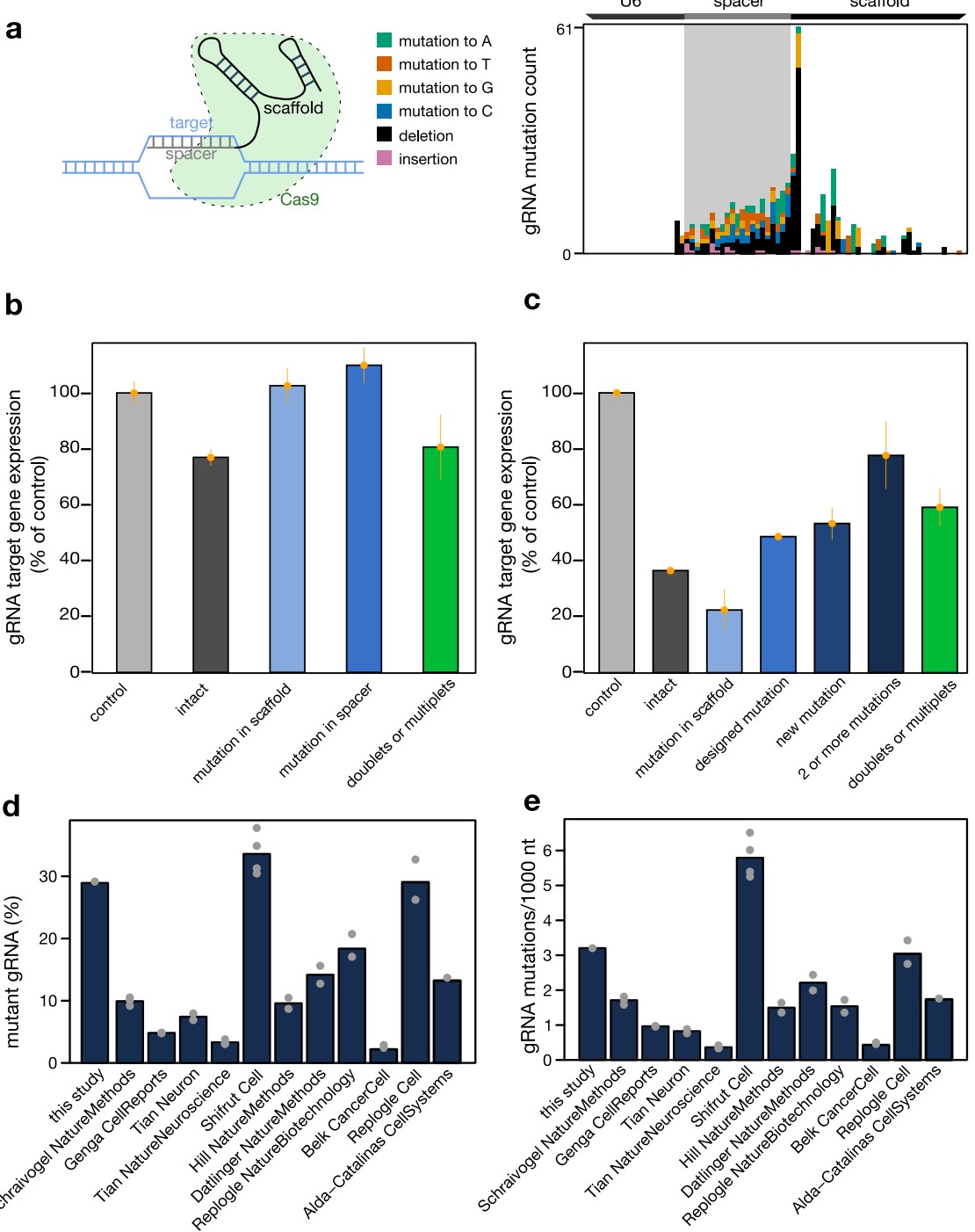

**Fig. 2 Application of GiRAFR to an extended dataset. a** Mutation frequency of gRNAs associated with cells after the Gaussian Mixture model filtering. The cartoon below illustrates the annotation of spacer and scaffold in the gRNA. **b** Average expression (log-transformed read counts) ± standard error of mean of gRNA target genes, normalized to their expression in cells with non-targeting gRNAs (control cells). 'intact' bar shows the expression in cells with intact gRNAs. 'mutation in scaffold', 'mutation in spacer' show the novel mutations identified by GiRAFR with mutations in the spacer region and gRNA spacer region. 'doublets or multiplets' show the cells with more than 1 gRNA newly identified by GiRAFR. **c** Similar as **b** using data from Jost et al.[14]. 'designed mutation' bar shows the expression of single mismatch gRNAs as designed in the experiment. 'mutation in scaffold', 'new mutation' and '2 or more mutations' show the novel mutations identified by GiRAFR with 0, 1, and 2 or more mutations in the spacer region. **d** Fraction of cells with a gRNA showing a mutation, out of all cells. Only gRNAs associated with cells after the Gaussian Mixture model filtering were considered. **e** Frequency of mutations in gRNAs associated with cells after the Gaussian Mixture model filtering. The mutation frequency is shown per cell and per 1000 nucleotides, to subtract differences in gRNA sequencing read length.

profiled (Fig. 2e, Supplementary Fig. 2a). Frequencies were higher in the spacer region and in regions immediately flanking it, corresponding to the sequences that are typically synthesized as pooled oligonucleotides for generating gRNA virus libraries

(Fig. 3a). Mutations, in addition, abound in the beginning and at the end of the spacer region—sites that serve as handles for cloning these oligonucleotides into the gRNA expression plasmid prior to virus production.

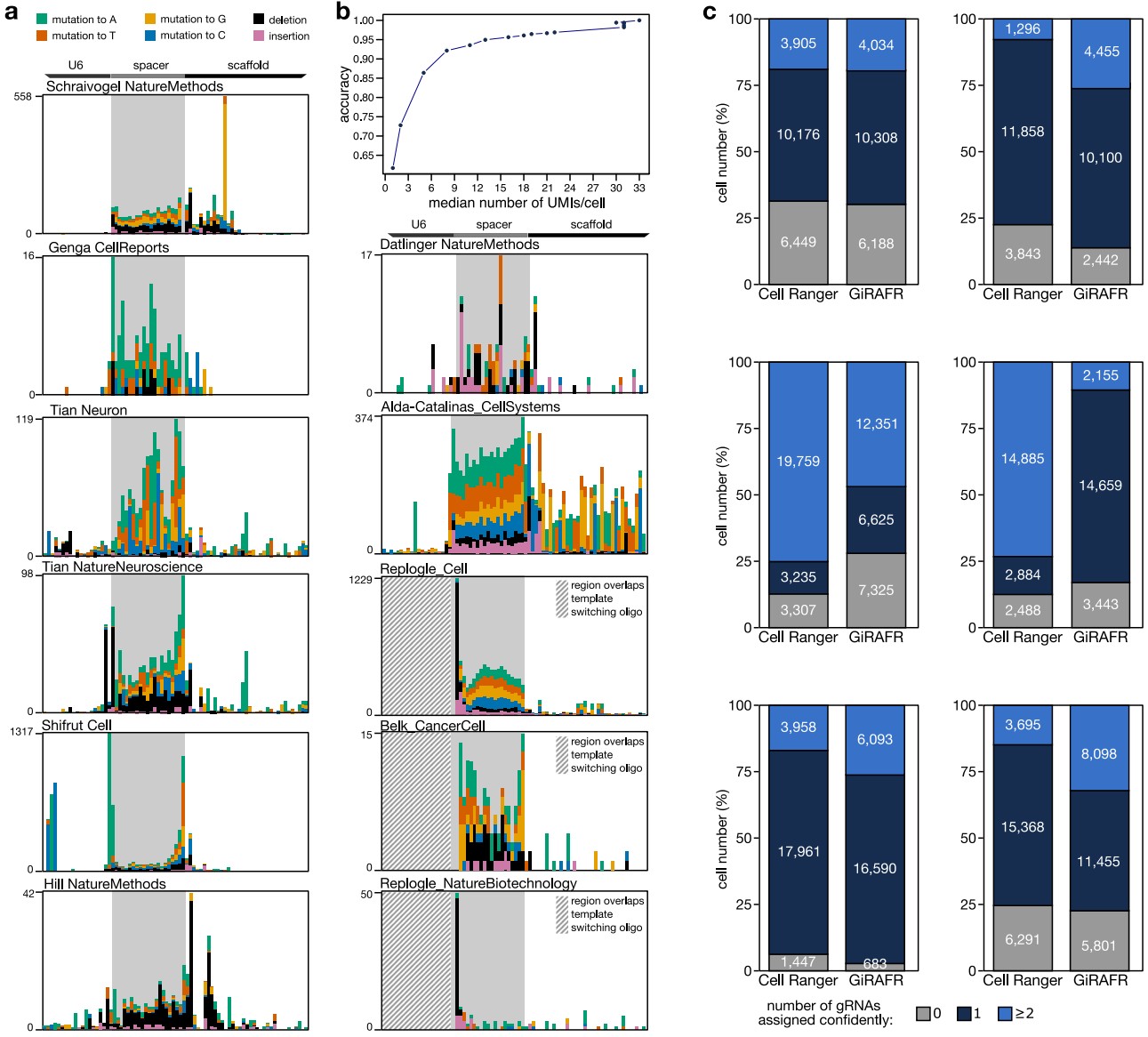

**Fig. 3 Mutation spectra of the extended dataset and cell assignment results of selected high sequencing depth experiments. a** gRNA mutation patterns detected in publicly available datasets, as in panel a. Shown is the aggregate across all experiments analyzed per dataset. Note that some gRNA libraries encompass only a fraction of the U6 promoter shown. Individual experiments are shown in Supplementary Fig. 2. Striped boxes indicate the position of template-switching oligonucleotides, where mutations on position −2, −1, and 0 before the start of the spacer were also removed. **b** gRNA assignment accuracy as a function of the median number of UMIs per cell, as estimated by downsampling the CROP-seq data generated in the current study. **c** Comparison of the number of different gRNAs assigned to a cell by Cell Ranger and GiRAFR. Shown are (left and right) data from studies Belk et al.[13] and Replogle et al.[3] (top), Tian et al.[11] and Genga et al.[8] (middle), and Schraivogel et al.[10] and Replogle et al.[14] (bottom).

**gRNA expression and sequencing saturation impact mutation calling**. In analyzing these 11 studies, we observed high variability in gRNA expression and in library sequencing saturation, factors likely to impact gRNA mutation calling accuracy. gRNA library sequencing saturation differed dramatically, ranging from 20 to 99%. Also, the average number of gRNA molecules detected per cell was highly variable, ranging from 1 to over 100 UMIs (Supplementary Fig. 2b). We first assessed the impact of this variation on how accurate gRNAs can be assigned to cells, by downsampling our own CROP-seq experiment. This analysis revealed that, at a sequencing saturation of 34, 59 and 92%, an average of 5, 16, and 31 unique gRNA molecules were recovered per cell, leading to a correct annotation of cells with their gRNA at an estimated 80%, 95% and 98% of samples (Fig. 3b). Cells may be incorrectly annotated for three reasons. First and foremost, low

sequencing depth will cause some fragments to be simply not detected. Second, gRNAs with low UMI counts can fail to meet the UMI count cutoff. And third, ambient gRNAs can be erroneous annotated as expressed as the automatic detection threshold shifts to a lower value. This number of inaccurate assignments upon downsampling is visualized in Supplementary Fig. 3a. Note that a low sequencing saturation also impacts mutation calling. Here, most UMIs are supported by 0 or 1 read and thus not included for analysis. As a result, fewer UMIs (and hence fewer intact as well as mutant gRNAs) are detected at lower sequencing depths (Supplementary Fig. 3b). In 13 of 26 experiments we analyzed, on average over 16 unique gRNA molecules were recovered per cell, suggesting that over 95% of cells are correctly assigned. We focused our analyses on these 13 experiments (6 studies); in the 13 other experiments, many cells were

excluded from analysis by GiRAFR as gRNAs cannot be assigned to cells with confidence.

**Validation of functional impact of gRNA mutations**. In each of the 13 high-depth experiments, GiRAFR identified more cells as containing a mutant gRNA (Supplementary Fig. 3c). Note that a head-to-head comparison between the analyses presented in each of these studies and GiRAFR is not feasible, as not each of them implemented the same UMI thresholding, and they often use ad hoc, fixed thresholds. To enable a head-to-head comparison, we therefore reanalyzed each of these datasets using Cell Ranger as well as GiRAFR. This revealed that Cell Ranger labeled between 6 and 40% of all cells as containing no gRNA, but 0.5 to 55% of these in fact represent cells transfected with mutant gRNAs as per our GiRAFR analysis. Likewise, cells labeled as having a single gRNA by Cell Ranger were often labeled as multiplets by GiR-AFR, with for example about one-third of single cells in one lane profiled by Replogle et al.[14] being mislabeled by Cell Ranger (Fig. 3c). As expected, mutant gRNAs showed attenuated functionality, having less impact on the expression of their target genes (Supplementary Fig. 4a). Similarly, multiplets mislabeled as singlets by Cell Ranger show a reduced downregulation of the target genes of the intact gRNAs compared to true singlets. Together, these analyses demonstrate the need for accurate and tailored gRNA detection, and confirm that GiRAFR outperforms Cell Ranger for gRNA library analysis and for assigning gRNAs to cells. Notably, we observed that the analysis runtime of GiRAFR scales in a linear manner with the product of the number of cells and the number of reads in the gRNA library (Supplementary Fig. 4b), taking about half a day to analyze 100,000 cells.

## Discussion

Single-cell CRISPR screening is increasingly being used to assess the impact of gene perturbations on cellular transcriptomes. Here, we developed a method to identify and annotate the gRNAs used in these assays more accurately, in a mutation-aware manner. GiRAFR takes alignment results (bam files) as its main input. It is compatible with the alignment outputs of both Cell Ranger and Dropseq, and should be compatible with other alignment files. While GiRAFR is clearly not essential for performing single-cell CRISPR screening (several studies have been performed without it), GiRAFR does stand to advance this burgeoning field by fixing several analysis gaps.

First and foremost, if mutant gRNAs cannot be detected, cells with a single gRNA cannot be properly distinguished from those with multiple gRNAs. Discriminating both sets of cells is important to avoid including cells that are inadvertently captured together and labeled with the same barcode. In that scenario, the transcriptome no longer represents the associated perturbation. We demonstrate that up to 22% of all cells profiled in earlier studies that are labeled as singlets by Cell Ranger are in fact multiplets, i.e., multiple cells or single cells transformed with 2 or more gRNAs. We also show that these undetected multiplets have no or an attenuated phenotype and that they can thus cloud downstream analyses.

Additionally, studies often use ad hoc customized pipelines and fixed thresholds for analysis. This renders a head-to-head comparison between each of these studies unfeasible. By applying a uniform analysis, we highlight strengths and weaknesses in several of these studies that can be taken into consideration when designing future experiments. For example, specific defined read depths need to be attained for accurate gRNA assignment and mutation detection.

Thirdly, GiRAFR can help to assess and optimize the quality of the experiment. The UMI used in gRNA sequencing allows for

discrimination between sequencing errors and mutations. GiR-AFR analyses can thus identify gRNA mutations and their sources, hence allowing research teams to verify the quality of the experimental work and remedy issues arising. Here, inspecting the mutation pattern along the gRNA sequence revealed that most mutations are spread across the gRNA spacer region, corresponding to the synthesized oligonucleotide fragment. This suggests that these mutations result from inaccurate oligonu-cleotide synthesis, and highlight a potential tradeoff between cheaper synthesis at lower accuracy, and the downstream infor-mation loss due to inaccurate oligonucleotide synthesis. In addition, the multiplicity of infection, the number of designed gRNAs and the mutations inadvertently introduced in them, and the number of cells per perturbation are all design choices that influence the ability to accurately assign gRNAs to cells. These choices can be optimized during the execution of the experiment, or remedied in part in analysis by finetuning parameters such as the minimal number of reads per UMI, the use of static or dynamic UMI thresholding, and deciding whether or not to pool mutant gRNA molecules.

Furthermore, GiRAFR-identified mutations allow for an ana-lysis of the impact of mutations on gRNA functioning. We for example demonstrate here that mutations in both the spacer and the scaffold compromise gRNA activity, as both are associated with a reduced ability to downregulate target gene expression. This is in line with an earlier study demonstrating that gRNA spacer mutations dampen perturbation phenotypes. We reported a high frequency of gRNA mutations in 11 published studies and demonstrated attenuated gRNA functionality in 5 of them. Worryingly, gRNAs with a single mutation (1 Hamming dis-tance) are labeled as intact by CellRanger but similarly show an attenuated phenotype. Although beyond the scope of the current study, more in-depth analysis of high-throughput datasets may enable a more fine-grained appraisal of which mutation types and locations are tolerable or damaging.

Finally, we demonstrate that both the number of gRNA UMIs identified per cell as well as the gRNA library sequencing depth affect the ability to detect gRNAs mutations and to reliably assign gRNAs to cells. Our data suggest that a high sequencing satura-tion facilitates identifying gRNA mutations. But importantly, the average number of gRNA UMIs differed dramatically between experiments. This may either be due to low gRNA expression or low detection rates, but can have a detrimental impact on the accuracy of assigning gRNAs to cells, discriminating ambient from endogenously expressed gRNAs, and singlets from multiplets.

Together, we believe that these notions support the need for accurate gRNA calling as implemented in GiRAFR, and that this novel software tool will therefore enhance the outcome of emergent and expanding single-cell CRISPR screens.

## Methods

**gRNA design**. We designed four different gRNAs for each of 25 target genes, and 20 non-targeting guides as controls using Benchling [Biology Software, 2018], retrieved from https://benchling.com. Final gRNA sequences contained homology arms (5′-end: TGGAAAGGACGAAACACCG, 3′-end: GTTTT AGAGCTAGAAATAGCAAGTTAAAATAAGGC) to allow for cloning into the CROP-seq-Guide-Puro vector (addgene plasmid #86708 from Christoph Bock). The gRNA library was synthesized and ordered as an oligo pool through CustomArray (GenScript).

**gRNA cloning and lentivirus productions**. Cloning of the pooled gRNA library, and all consecutive steps until library preparation, were performed according to the CRISPR droplet

sequencing (CROP-seq) protocol as detailed[4]. Briefly, the vector was digested with 20 units of *BsmBI* (NEB, R0580L), and the backbone was purified using the SNAP UV-Free Gel Purification kit (Fisher Scientific, 45-0105). The gRNA library pool was cloned into the resulting backbone fragment with NEBuilder HiFi DNA Assembly Master Mix (NEB cat no. E2621S) and transformed into Endura Electrocompetent cells (Lucigen cat. No. 60242). Plasmids were isolated and purified using the QIAprep Spin Miniprep Kit (Qiagen, 27104). This plasmid pool was subsequently transfected in HEK293T cells together with pMDLg/pRRE, pRSV-Rev, and pMD2.G (Addgene #12251, #12253, #12259) using lipofectamine 3000 (ThermoFisher, L3000015), to produce lentivirus. For lentiviral titration, puromycin (Cayman Chemicals, 13884) resistant colonies were counted through Crystal Violet (Sigma Aldrich, 61135) staining as per the manufacturer's instructions.

**Generation of A549-TetOn-Cas9 cells**. A549 cells (ATCC) were transduced with a lentivirus encapsulating the pCW-Cas9-Blast vector (Addgene, #83481), followed by selection with 20 µg/mL blasticidin (InvivoGen, 38220000) for 14 days. Cells were validated as being successfully transformed by western blotting for Cas9.

**CROP-seq**. The A549-TetOn-Cas9 cells were transduced with the lentiviral gRNA library pool at a multiplicity of infection of 0.3. Cells were grown in DMEM high glucose (ThermoFisher, 41965062) at 37 °C in 5% CO2 and passaged every 2 days with Trypsin-EDTA (ThermoFisher, 25200056). Positively transduced cells were selected for puromycin (100 µg/mL). Following selection for 14 days, Cas9 expression was initiated by doxycycline (VWR, J60579.14) addition (5 µg/mL). After two days of induction, cells were grown for another 5 days, collected, and processed following the 10x Genomics demonstrated protocol for the preparation of single-cell suspensions. ~5000 cells were processed with Chromium Next GEM Single Cell 3′ GEM, Library & Gel Bead v3.1 kit (10X Genomics, 1000128) following the manufacturer's instructions.

10% of cDNA was used for gRNA enrichment and the creation of a specific gRNA library. For higher resolution and better cell assignments, gRNA sequences were amplified from the transcriptome library before the last indexing PCR with Hifi HotStart ReadyMix (Roche, 7958935001) and gRNA cassette specific primers (5′-CAAGCAGAAG ACGGCATACG AGATXXXXXXXXXXGT GACTGG AGTTCAGACG TGTGCTCTTC CGATCTTCTT GTGG AAAGGA CGAAACACCG-3′ and 5′-AATGATACGG CGACCACCGA GATCTACACT CTTTCCCTAC ACGACGCTCT TCCGATCT-3′, both at a final concentration 0,5 µM) in a thermal cycler (Westburg, AJ8462070241). The final transcriptome and gRNA libraries were purified using Ampure XP beads (Analis, A63881) and analyzed using a Bioanalyzer high sensitivity DNA analysis kit (Agilent, 5067-4626), and sequenced using a NovaSeq 6000 (Illumina, 20068232), with 28 and 91 nucleotides from read 1 and 2, respectively. We obtained respectively 454,191,138 and 25,289,660 reads from the transcriptome and gRNA libraries.

**CROP-seq data processing using cell ranger**. Alignment of the gene expression library, detection of spacer, and cell assignments were performed by Cell Ranger (version 3.1.0) feature barcoding analysis. The count matrix of cells with a single gRNA spacer was then extracted and normalized using LogNormalize function (R version 4.2.0[18] and R package Seurat, version 4.1.0[19]) with a scale factor 10,000 for further analysis.

**CROP-seq data processing using GiRAFR**. GiRAFR relies on mapping of reads to a custom reference genome, filtering of mapped reads, generation of a consensus call, and identification of mutuations therein. Below, each of these steps is explained in more detail.

*Generation of custom reference genome*. A custom reference genome was built using Cell Ranger mkref, a build note can be found in our user manual. It was composed of the human reference genome GRCh38, supplemented with an artificial chromosome for each gRNA, containing the gRNA spacer sequence as well as upstream and downstream sequences relevant to the underlying method. Specifically, for data using 10X protocol, we used 530 basepairs from the plasmid containing the gRNA cassette. For data originating from Dropseq, we included the requisite Dropseq-tools index in the custom reference genome file.

*Mapping of reads to the custom reference genome and filtering of mapped reads*. Next, reads from the targeted sequencing library and the transcriptome sequencing library were aligned to this custom reference genome by STAR as implemented in Cell Ranger or Dropseq-tools. For 10x sequencing, Cell Ranger performed correction on the cell barcode and UMI with a maximal 1 hamming distance. All subsequent analyses were limited to reads having both a valid cell barcode and a valid UMI (CB and UB tag, or XC and XM tag for 10x and Dropseq respectively). After the removal of the secondary alignments, reads mapping successfully to the artificial gRNA references were collected in a BAM file.

This bam file was filtered by the predefined cell barcodes list, containing cell barcodes that remain after the removal of background noise. In most cases, this list came from alignments of gene expression libraries. To solve cell barcode discrepancies between gene expression libraries and captured CRISPR libraries by 10X feature barcoding technology, unmatched cell barcodes were compared and corrected with the lookup table provided by Cell Ranger.

*gRNA consensus sequence calling*. Part of the reads in the gRNA-filtered BAM file are associated with an identical UMI and cell barcode. These are assumed to originate from the same cDNA molecule. Because of errors in library preparation and/or sequencing, the associated nucleotide sequence in these reads is not necessarily identical. We therefore defined a consensus sequence for each UMI-cell barcode combination. Specifically, the Pysam (https://github.com/pysam-developers/pysam) (Python version 0.15.3) package[20] was used to extract the set of reads for UMI-cell barcode combination from the gRNA filtered BAM file. Firstly, the mapped gRNAs of this set of reads were inspected (GN tag for cell ranger and gn for dropseq-tools bam file). In most cases, they align uniquely to one gRNA. In the case of different gRNA annotations for reads with the same UMI-cell barcode combination, the gRNA annotation supported by the highest number of reads was kept. In the (unlikely) event of a tie, no gRNA can be confidently mapped, and reads were kept out from the subsequent gRNA calling steps. From the subset of reads that unanimously mapped, the common sequence supported by the most reads was defined as the consensus sequence for this UMI-cell barcode combination. When multiple different sequences have identical reads counts, we randomly selected one as the consensus sequence.

*Filtering of gRNA consensus sequences*. In theory, those consensus sequences from UMI-cell barcode combinations that were constructed using multiple reads, can be considered as of higher confidence, because this excludes sequencing errors. In GiRAFR,

we provide the option to filter consensus sequences according to the number of reads supporting them (default: >1 read). Cell barcodes, UMIs, and corresponding consensus sequences were written into "consensus.sequence.gRNA.txt" file. These alignments were also saved into a "consensus.bam" BAM file which is compatible with other analysis tools.

*Identification of mutations in the gRNA consensus.* Analyzing the alignment information is a fast and accurate way to identify mutations in short sequences such as in the oligonucleotides used to construct gRNA libraries. With the designed gRNA spacer and cassette sequence annotation, we uncover changes from the consensus sequences we constructed per base. From the CIGAR strings in BAM alignments files, GiRAFR keeps all insertions (I), deletions (D), skipped regions (N), and soft clippings (S) and compares mapped bases to references (M). GiRAFR can annotate the mutation location onto the oligo structure annotation provided by the user and detailed mutations in a manner similar to CIGAR (details in Supplementary Note 1). Perfect matching without any mismatch will be defined as an intact gRNA molecule. Variant gRNAs were named with number suffixes.

*Assign gRNAs to cells.* Ambient gRNAs molecules can be captured in droplets and erroneously assigned to cells. To avoid this, a threshold defining the minimal number of UMIs supporting the assignment of a gRNA to a cell is defined. Two modes are available. Users can define a fixed threshold (for example three UMIs) which will be applied to all gRNAs. Only gRNAs in a cell with more UMIs than the threshold will be assigned to that cell. Alternatively, automatic dynamic detection of UMI thresholds can be employed. It will fit a two-component mixture Gaussian model to model two distributions of in-cell gRNA UMIs and background noise ("ambient") UMIs. From these two distributions, the minimum number of UMIs in the gRNA distribution is set as the UMI threshold for that gRNA. Intact gRNA and their respective variants are counted together in fitting this distribution. This method is inspired by and adopted from Cell Ranger. The resulting output consists of a matrix containing the UMI counts per guide (including mutated guide) and per cell.

*Detect CRISPR-cas9 editing effect.* Alignments from the expression library are first split into sub bam files to accelerate detection speeds by SAMtools (version 1.9)[21]. Each sub bam file contains the alignments that mapped over the detection window of each gRNA. The detection window is short and symmetric around the Cas9 nuclease cutsite on the genome, which is 3–4 nucleotides upstream of the PAM site. We used a 51 bp detection window. The FeatureCounts (v2.0.1)[22] tool was used to annotate mapped reads to genes. Only alignments that are assigned and mapped to target genes are considered valid.

Similarly, to deduplicate reads with identical UMI and cell barcode, we construct a consensus sequence. To address that reads do not align to the same starts within the detection window, we take unions of sequenced reads as the consensus sequence for each UMI-cell barcode combination. By analyzing mapped positions of consensus sequences, if deletions happen across the cutsite, they are detected as editing effects. Insertions or deletion elsewhere and mismatches are not considered as "Cas9-induced", but are also reported in the final outputs.

**Statistics and reproducibility.** The downregulated expression of 14 target genes in KO cells ($n = 1655$) compared to the expression in Control cells ($n = 10,598$) was tested by two-sided $t$ test provided by R package: ggpubr (version 0.4.0.999)[23]. The expression of target genes in cells with intact gRNAs ($n = 2004$), gRNAs with a mutant

spacer ($n = 452$) and gRNAs with a mutant scaffold ($n = 380$) were compared to those in control cells ($n = 18,925$) and were tested by welch two-sample $t$ test using R[18].

**Reporting summary.** Further information on research design is available in the Nature Portfolio Reporting Summary linked to this article.

## Data availability
CROP-seq data for this study has been deposited in the NCBI's Gene Expression Omnibus (GEO) database under the accession number GSE216040. Processed data can be download by figshare: https://doi.org/10.6084/m9.figshare.24064854. All other data are available from the corresponding author on reasonable request.

## Code availability
GiRAFR is available as an open-source Python package at our GitHub Repository (github.com/FunctionalEpigeneticsLab/GiRAFR) and Zenodo repository with https://doi.org/10.5281/zenodo.833371[24]. Scripts to recreate all figures using processed data are also included in this repository as jupyter notebooks.

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

## Acknowledgements

We thank all authors of the data used in this study for their previous work and for making their results available. We thank the Leuven Genomics Core for sequencing and Vlaams Supercomputer Center for computing resources. This work was supported by funding through the host institution, KU Leuven (C1 grant to B.T.), as well as external grants; Foundation against Cancer (F/2020/1544) and Flanders (FWO), grant nos. 11K6222N (to P.V.M.). Funding for open access charge: Foundation against Cancer.

## Author contributions

Q.Y. wrote the software as well as ran the analyses detailed in this manuscript. Q.Y. and B.T. conceived the project. Q.Y. and B.T. wrote the manuscript with contributions from P.V.M. E.G. performed experiments for the In-house screen. P.V.M. provided feedback on the interpretation of the results. All authors read and approved the final manuscript.

## Competing interests

B.T. holds stock for 10X Genomics. The remaining authors declare no competing interests.
