## [Peer Review File · Communications Biology]

GiRAFR improves gRNA detection and annotation in single cell CRISPR screensReviewers' comments:

Reviewer #1 (Remarks to the Author):

Yu et al present a software that corrects gRNA mutations in crop-seq CRISPR experiments. The method is novel, the paper is easy to read, the figures are clear. I'm not an expert of CROP-seq assays, but given how common single cell and bulk sequencing barcode mutations are, it makes sense that they also complicate crop-seq assays, and need to be addressed in the analysis. Not knowing this assay well, the only thing I'm surprised about is why this type of correction doesn't exist already.

Minor points:

Software: I wasn't able to install the software at first. The authors should point out that one has to be running Python 3.8 to be able to run the pip install command they provide. Otherwise, the pip command is great.

Typo:

- > Ambient gRNAs molecules libraries can be captured in droplets and to
- > erroneously assigned to cells.

Reviewer #2 (Remarks to the Author):

This study emphasizes the importance of identifying and annotating dysfunctional mutant gRNAs in single cell CRISPR screens to improve the quality of the analysis. The researchers developed the GiRAFR tool to address the lack of dedicated software for precise annotation and association of these disfunctional CRISPR gRNA libraries with single cell transcriptomes.

Overall, this paper presents a valuable solution by introducing GiRAFR as a dedicated tool for annotating CRISPR gRNA libraries. I have no further questions and would like to accept this well-written paper.

Reviewer #3 (Remarks to the Author):

This is a review of the study reported by Yu et al.

The study presents a new tool named GiRAFR (Guide RNA Anomaly and Functionality Revealer) aimed at improving the analysis and quality control of single cell CRISPR screens. It addresses a recurring issue of undetected mutant guide RNAs (gRNAs) which can lead to inaccuracies in cell annotations and estimates of genetic perturbations. GiRAFR was designed to identify and characterize both intact and mutant gRNAs, and its application demonstrated that such mutant gRNAs are common in libraries and can cause substantial inaccuracies in data analysis.

While I greatly appreciate the development of open-source tools, such as GiRAFR, which presents a great contribution to the research community in the field of single-cell genomics and CRISPR screens, I think the manuscript should further address a few aspects of the tool's performance and as well as that of data interpretation.

The major comments that need to be addressed:

1. While the presented approach to detect mutants is clever and does offer flexibility in calling thresholds, I do not fully agree with the interpretation of mutant origins. The authors state that 'GiRAFR enables profiling of gRNA sequence variations, as well as allowing to pinpoint the sources of

this variation, such as induced during library preparation or during virus preparation.’
However, this capability is not demonstrated in the results, neither it is clearly supported by the figures. The authors should explain how GiRAFR corrects for the library prep and sequencing errors which can occur within gRNA.

2. The Results section ‘gRNA expression and sequencing saturation impact mutation calling’ requires further clarification. It is unclear why lower copies per cell would yield inaccurate cell assignment. Is it the cell barcode that can no longer be matched correctly or gRNAs are cut off by the threshold? Similarly, it is unclear why sequencing saturation would impact mutation calling. Please explain and demonstrate using data.

3. Please provide a textual explanation of the tool's scalability, the robustness of its performance across different experimental conditions, as well as the anticipated mode of usage (with Cell Ranger or independent of it).

Minor noted discrepancies:

Typos in line 300

‘Mutations in addition abound in the beginning and at the end of the spacer region, sites that serve as handles for cloning these oligonucleotides into the gRNA expression plasmid prior to virus production.’

Reviewers' comments:

Reviewer #1 (Remarks to the Author):

Yu et al present a software that corrects gRNA mutations in crop-seq CRISPR experiments. The method is novel, the paper is easy to read, the figures are clear. I'm not an expert of CROP-seq assays, but given how common single cell and bulk sequencing barcode mutations are, it makes sense that they also complicate crop-seq assays, and need to be addressed in the analysis. Not knowing this assay well, the only thing I'm surprised about is why this type of correction doesn't exist already.

Minor points:

Software: I wasn't able to install the software at first. The authors should point out that one has to be running Python 3.8 to be able to run the pip install command they provide. Otherwise, the pip command is great.

We thank the reviewer for this comment. We have released a new version and tested it with Python 3.7.6. We also added dependencies description to the online website: <https://girafr.readthedocs.io/>

Typo:

Ambient gRNAs molecules can be captured in droplets and to erroneously assigned to cells. This has now been corrected.

Reviewer #2 (Remarks to the Author):

This study emphasizes the importance of identifying and annotating dysfunctional mutant gRNAs in single cell CRISPR screens to improve the quality of the analysis. The researchers developed the GiRAFR tool to address the lack of dedicated software for precise annotation and association of these dysfunctional CRISPR gRNA libraries with single cell transcriptomes.

Overall, this paper presents a valuable solution by introducing GiRAFR as a dedicated tool for annotating CRISPR gRNA libraries. I have no further questions and would like to accept this well-written paper.

We thank the reviewer for taking the time to review our paper and for the positive feedback.

Reviewer #3 (Remarks to the Author):

This is a review of the study reported by Yu et al.

The study presents a new tool named GiRAFR (Guide RNA Anomaly and Functionality Revealer) aimed at improving the analysis and quality control of single cell CRISPR screens. It addresses a recurring issue of undetected mutant guide RNAs (gRNAs) which can lead to inaccuracies in cell annotations and estimates of genetic perturbations. GiRAFR was designed to identify and characterize both intact and mutant gRNAs, and its application demonstrated that such mutant gRNAs are common in libraries and can cause substantial inaccuracies in data analysis.

While I greatly appreciate the development of open-source tools, such as GiRAFR, which presents a great contribution to the research community in the field of single-cell genomics and CRISPR screens, I think the manuscript should further address a few aspects of the tool's performance and as well as that of data interpretation.

The major comments that need to be addressed:

1. While the presented approach to detect mutants is clever and does offer flexibility in calling thresholds, I do not fully agree with the interpretation of mutant origins. The authors state that 'GiRAFR enables profiling of gRNA sequence variations, as well as allowing to pinpoint the sources of this variation, such as induced during library preparation or during virus preparation.' However, this capability is not demonstrated in the results, neither it is clearly supported by the figures. The authors should explain how GiRAFR corrects for the library prep and sequencing errors which can occur within gRNA.

We thank the reviewer for this comment, and for highlighting this unclear aspect. The GiRAFR output provides mutation annotation for each UMI in the gRNA library, which allows further investigation of the source of variation. To demonstrate this, we have annotated the mutations from our in-house CROP-seq gRNA library. We distinguished 5 scenarios as sources of mutations (from line 235 to 258, Figure S1f). In the revised manuscript, we now go through these 5 scenarios in detail, and demonstrate how gRNA mutations can be assigned to (subsets of) these scenarios.

Revised manuscript (line 235 to 258) and figures:

Potential sources of gRNA mutations

We next sought to further analyze the sources of these mutant gRNAs, using the GiRAFR output (Figure S1f). Most UMIs that associate with a mutant gRNA (97.3 %) originated from 445 unique mutant gRNAs. These mutant gRNAs were each detected in multiple cells (3,173 cells in total). They were thus most likely already amplified in the virus pool used for transduction. We cannot discriminate between those originating from inaccurate oligonucleotide synthesis and from errors introduced during gRNA cloning (Figure S1f, scenario 1). A smaller number of mutant gRNAs only appeared in one cell but were supported by all UMIs for that gRNA in that cell ($n = 23$ gRNAs, 23 cells). These mutations are likely derived from errors during lentivirus transduction, but they could equally represent a rarer mutant gRNA sub-clone present in the virus pool (Figure S1f, scenario 1 or 2). Interestingly, in 1,100 cells, both the mutant and intact version of the same gRNA were encountered. In 1,062 of these 1,100 cells, only one single UMI supported the presence of the mutant gRNA (validated across multiple reads) while one average 56 UMIs supported the presence

of the intact gRNA. Here, the mutation most likely occurred in cDNA preparation (Figure S1f, scenario 4). This contrasts with the cells where more than 1 UMI supported detection of both the mutant and intact gRNA ($n = 78$). Under the latter constellation, the mutation was most likely introduced during lentiviral integration, which is associated with a duplication of the gRNA expression cassette (4). Mutations introduced during this integration/duplication will give rise to an intact and a mutant gRNA both being expressed in a single cell (Figure S1f, scenario 3). Note that 40 cells contain different gRNA mutations, either supported by 1 UMI and by >1 UMI. In a final scenario, a mutation is detected in a single read of a UMI, but not in the other reads of that UMI. This is a sequencing error (Figure S1f, scenario 5) which, as described higher, is filtered out by GiRAFR during the consensus sequence generation step. Here, 27 % of the raw mutant molecules were filtered out and 91% of the consensus sequences were constructed from sequence variations.

Figure S1f: Schematic overview of CROP-seq experiment with different sources of gRNA mutations. ①: Inaccurate oligonucleotide synthesis ②: Errors during lentivirus transduction ③: gRNA cassette genome integration ④: Errors induced during gRNA transcription or during cDNA synthesis ⑤: Sequencing errors

Regarding the request to explain how GiRAFR corrects for sequencing errors, the process of consensus read calling was explained in Figure 1d and is now described in more detail in line 222-226.

Revised manuscript (line 222 to 226):

Specifically, it first calls a consensus gRNA sequence for each UMI. In doing so, sequencing errors can be filtered from the gRNA pool as detection of multiple reads supporting the same gRNA sequence becomes a prerequisite for gRNA mutation calling (Figure 1d). Next, it generates a count matrix per detected gRNA, whilst discarding UMIs with fewer than 2 reads to avoid including sequencing errors as consensus gRNAs.

2. The Results section ‘gRNA expression and sequencing saturation impact mutation calling’ requires further clarification. It is unclear why lower copies per cell would yield inaccurate cell assignment. Is it the cell barcode that can no longer be matched correctly or gRNAs are cut off by the threshold? Similarly, it is unclear why sequencing saturation would impact mutation calling. Please explain and demonstrate using data.

We thank the reviewer for this comment. As he/she likely knows, a major issue in scRNA-seq analyses is ambient RNA, i.e. RNA molecules released by (dead) cells that are captured together with an intact cell in a droplet. If only a few gRNA copies are detected per cell,

discriminating ambient from cell-derived gRNAs can be problematic (e.g. for both, a single copy is detected). Only when the gRNA is highly expressed and multiple UMIs are detected per gRNA, can we discriminate ambient gRNAs (a single molecule is detected of a given gRNA species) from cell-associated gRNAs (dozens of molecules are detected of a single gRNA species). To address this in more detail, we down-sampled our inhouse CROP-seq dataset, and compared which gRNA is assigned to each cell following low-depth sequencing, to assignment upon higher-depth sequencing. As the sequencing depth is lowered, we observed an increase in the number of cells where no gRNA could be detected (grey bar), that have the wrong gRNA assigned to them (green bar), or where not all expressed gRNAs were detected (blue bar) (from line 330 to line 337, Figure S3a).

We leveraged the same down-sampling to visualize how low sequencing depths affect gRNA mutation calling. Briefly, given our consensus calling approach, we require at least 2 reads of the same UMI, for that UMI (and its gRNA sequence) to be reliably called. Hence, at low sequencing depths, most UMIs are supported by 0 or 1 read and not included in the analysis. As a result, not only intact gRNAs are detected at a lower rate, but also a lower number of mutant UMIs (and hence mutant gRNAs) are detected at lower sequencing depths (Figure S3b).

Revised manuscript (line 330 to 337) and figures:

Cells may be incorrectly annotated for 3 reasons. First and foremost, low sequencing depth will cause some fragments to be simply not detected. Second, gRNAs with a low UMI counts can fail to meet the UMI count cutoff. And third, ambient gRNAs can be erroneous annotated as expressed as the automatic detection threshold shifts to a lower value. This number of inaccurate assignments upon down-sampling is visualized in Figure S3a. Note that a low sequencing saturation also impacts mutation calling. Here, most UMIs are supported by 0 or 1 read and thus not included for analysis. As a result, fewer UMIs (and hence fewer intact as well as mutant gRNAs) are detected at lower sequencing depths (Figure S3b).

Figure S3: a. Number of inaccurate assignments in down-sampled inhouse CROP-seq experiment. Compared to full data (5k reads per cell), mis-assignments are categorized as no gRNA could be detected (grey), not all expressed gRNAs were detected (blue), and wrong gRNA assigned (green). **b.** Number of detected UMIs in down-sampled inhouse CROP-seq experiment gRNA library.

3. Please provide a textual explanation of the tool's scalability, the robustness of its performance across different experimental conditions, as well as the anticipated mode of usage (with Cell Ranger or independent of it).

We thank the reviewer for this comment, which we have addressed both textually and quantitatively. To assess scalability, we determined how the runtime of GiRAFR is affected by the sequencing depth and by the number of cells in the gRNA library. This revealed that GiRAFR runtime scales in a linear manner with the product of the number of cells and the number of reads in the gRNA library, and this both when down-sampling our own inhouse data, and when looking across all datasets analyzed (Figure S4b, left and right panels, respectively). Of note, the largest dataset consisted of ~120,000 cells and required 15 hours run-time. This information was added to the manuscript (lines 358 to 360)

Revised manuscript (line 358 to 360) and figures:

Notably, we observed that the analysis runtime of GiRAFR scales in a linear manner with the product of the number of cells and the number of reads in the gRNA library (Figures S4b), taking about half a day to analyze 100,000 cells.

Figure S4b: Correlation between GiRAFR runtime and the product of cells number and reads number using inhouse down sample data (left) and publicly available single cell CRISPR experiments (right). Shown are Pearson correlations and runtime of input bam file cleaning and preparation was not included.

Anticipated mode of usage:

GiRAFR takes alignment results (bam files) as its main input. It has been tested to be compatible with the alignment outputs of the Cell Ranger and the Drop-seq algorithms. GiRAFR is also compatible to other alignment files by simply modifying its configuration (tag information). While most of our analyses have been done jointly with Cell Ranger, GiRAFR is hence not dependent on this mode of usage.

We have now added these descriptions to the manuscript (from line 365 to 367), as well as to the online website: <https://girafr.readthedocs.io/>

Revised manuscript (line 365 to 367):

GiRAFR takes alignment results (bam files) as its main input. It is compatible with the alignment outputs of both Cell Ranger and Drop-seq, and should be compatible with other alignment files.

Minor noted discrepancies:

Typos in line 300

'Mutations in addition abound in the beginning and at the end of the spacer region, sites that serve as handles for cloning these oligonucleotides into the gRNA expression plasmid prior to virus production.'

This has now been corrected.

REVIEWERS' COMMENTS:

Reviewer #3 (Remarks to the Author):

Thank you for addressing my comments. I appreciate the work done on the manuscript to address the suggested improvements. I am happy to recommend the updated manuscript for publication.